# Data Quality in Geochemical Elemental and Isotopic Analysis

**V. Balaram \***  **and M. Satyanarayanan** 

CSIR—National Geophysical Research Institute, Hyderabad 500 007, India
* Correspondence: balaram1951@yahoo.com

**Abstract:** Appropriate sampling, sample preparation, choosing the right analytical instrument, analytical methodology, and adopting proper data generation protocols are essential for generating data of the required quality for both basic and applied geochemical research studies. During the last decade, instrumental advancements, in particular further developments in ICP-MS, such as the use of tandem ICP-MS, high-resolution mass spectrometry to resolve several interferences, and the use of the second path with a collision/reaction cell in multi-collector ICP-MS (MC-ICP-MS) to effectively resolve interferences, have brought in remarkable improvements in accuracy and precision in both elemental and isotopic analyses. The availability of a number of well-characterized geological certified reference samples having both elemental and isotopic data-enabled matrix-matching calibrations and contributed to the quality and traceability of the geochemical data in several cases. There have been some developments in the sample dissolution methods also. A range of quality issues related to sampling, packaging and transport, powdering, dissolution, the application of suitable instrumental analytical techniques, calibration methods, accuracy, and precision are addressed which are helpful in geochemical studies.

**Keywords:** geochemical analysis; sample dissolution; calibration; element; isotope; geostatistics; data quality; accuracy; precision; reference materials

## 1. Introduction

Geochemistry is a constantly expanding science, and contemporary geochemical research has broadened into several sub-disciplines such as isotope geochemistry, exploration geochemistry, environmental geochemistry, and analytical geochemistry [1,2]. Currently, geochemical studies are playing a greater role not only in solving several mysteries of the Earth and the universe and finding out hidden mineral deposits, but also in understanding and resolving important environmental issues such as climate change, environmental pollution, and health. Geochemical studies also helped in solving many of the unresolved questions related to the Earth's earliest crust and crustal processes in the Precambrian era. For example, very recently some trace-element proxies of the tectonic-magmatic settings indicated evidence of Plate Tectonics in a 3.8-billion-year-old zircon crystal [3]. As a result, the data produced today in geochemical laboratories and the quality of analytical data have become more important for both basic and applied geochemical studies, and also for the decisions related to the environment and ecology. Information on the accurate chemical composition of various earth materials (e.g., bedrocks, minerals, soils, till, regolith, lake sediments, stream sediments, and water) is often fundamental to these investigations. In geochemical studies, major and minor elements data (e.g., $SiO_2$, $Al_2O_3$, and $MnO$) are expressed in wt.%, and trace elements (e.g., Sr, Cu, Co, and Ni) are usually expressed in μg/g or ng/g (*ppm* or *ppb*). Both basic and applied geochemical studies require data of not only major and minor elements, but also several groups of trace elements, such as large ion lithophile elements (LILE: e.g., K, Rb, Sr, Cs, and Ba), high field strength elements (HFSE: Hf, Zr, Ti, Nb, and Ta), rare earth elements (REE), platinum group elements (PGE), Au, U, and Th. In addition, age dating and isotope geochemical studies require highly precise isotope and isotopic ratio information on both stable and radiogenic isotopes in rocks, minerals,

sediments, and water. Geochemical baseline mapping containing the information on the spatial distribution of chemical elements at the Earth's surface was originally developed for its use in mineral exploration but now is finding wide applications in environmental sciences and health [4–7]. Requirements of element/isotope data for understanding various geochemical process including ore formation are presented in Table 1.

**Table 1.** Requirements of element/isotope data for understanding various geochemical processes including ore formation.

| Component in Rock | Utility |
|---|---|
| Whole rock—major and minor elements | Classification/discrimination, plotting variation diagrams, qualitative and quantitative modeling. Understanding redox conditions (Mn, Fe, etc.). |
| Trace elements—Ba, Rb, Sr, Zr, Hf, U, REE, Y, Sc, V, Ni, Mo, Pb, etc. | Diverse trace elements and their behavior help in understanding the wide range of geological and geochemical processes, petrogenesis, and ore-genesis. |
| REE, PGE, Ag, Au, Cu, Co, Ni, Zn, Li, etc. | Exploration, modeling |
| As, Bi, Sb, Hg, Se, Te, Pd, I, Ni, etc. | Exploration pathfinders, modeling |
| Elemental ratios: Mg/Fe, Cr/Al and Ca/Na, etc. | In mineral exploration to understand the chemical and mineralogical expressions of large- and local-scale process. |
| Radiogenic isotopes, $^{238}$U–$^{206}$Pb (4.47 by); $^{235}$U - $^{207}$Pb (710 my); $^{232}$Th -$^{208}$Pb (14.05 by); $^{87}$Rb–$^{87}$Sr (49.23 my), $^{40}$K–$^{40}$Ar (1.25 by), etc. | Natural variations in the relative abundances of isotopes of various elements provide information and clues for absolute dating, constraining magma sources, and processes such as crust-mantle interactions, and nuclear processes. |
| Stable isotopes: C, O, N, S, Mg, V, Cu, Mo, Ba, K, Pt, Pd, Ag, Ce, Er, Si, etc. | They provide information on physicochemical processes such as water-rock interactions, fluid/sulfur, carbon sources, crystallization temperatures, and fractionation by redox-related and/or biological processes. |
| Noble gases, and naturally occurring artificial isotopes such as $^{10}$Be, $^{14}$C, $^{20}$Al, $^{36}$Cl, and $^{129}$I which are formed by cosmic ray interaction in upper atmosphere | Modeling the rates of more recent geochemical processes: e.g., dating of geologically young materials such as soils, etc. Noble gas isotopes are used to understand the ore-forming processes |

A large number of laboratories worldwide produce millions of analytical data (both elemental and isotopic) related to both basic and applied geochemical studies which form the basis for proposing new theories/concepts in basic studies or starting an exploration or a mining project in a big way [8]. In such situations, the accuracy of analytical measurements is a prerequisite for sound decision-making. Hence, the analytical results obtained in a laboratory must be highly accurate and a true reflection of the sample analyzed. The purpose of this paper is to update the current trends in geochemical analysis, and provide basic information and guidance about developing quality assurance and quality control (QA and QC) protocols during geochemical analysis in geochemical laboratories for the generation of precise and accurate data required in both basic and applied geochemical studies.

## 2. Major Factors Responsible for the General Decline in Quality of the Analytical Data

A study conducted by Weis et al. [9] revealed that out of 7200 analytical geochemistry publications, only 9% presented the measurement of the results for certified reference materials (CRMs) which provide traceability. This means that the vast majority of the results that are being currently published are not validated, which is not a healthy situation. The development of quality control protocols for geochemical analysis depends largely on practices and procedures such as the appropriate use of CRMs during analytical pro-

tocols, regular participation in round-robin tests, international proficiency test trials, and interlaboratory comparisons [10,11]. Today, the vast majority of geochemical data are being obtained all over the world from commercial laboratories. While these laboratories are also accredited and have their own internal quality control (QC) and quality assurance (QA) procedures, a number of quality issues such as sample mix-ups, cross-contamination during transport, and several other problems can lead to the generation of erroneous data. As a result, on several occasions, the analytical results received from different laboratories, and even from the same laboratory at different times, are not necessarily comparable. The lack of competence of laboratory staff is believed to be mainly responsible for the large uncertainties observed normally in interlaboratory comparisons in quality assurance in the chemical analysis [12]. Hence, the importance of QA and QC are still serious issues.

### 3. Practice of Quality Assurance (QA) and Quality Control (QC) Protocols in Geochemical Analysis

QA and QC are the two fundamental requirements to ensure the reliability of analytical results. Quality assurance is expressed in terms of accuracy/precision and control expressed through the usage of blanks, standards, duplicates, and replicates. Data that are not based on adequate QA and QC can be erroneous, and the use of such data in geochemical investigations can lead to misleading conclusions. While QA aims at preventing problems, the purpose of QC is to detect them, and in the event that they occur, assess their extent, and take the appropriate actions to minimize their effects.

### 3.1. Target Precision ($\sigma$)

How accurate and precise the given data is depends on many factors, and in the real world, every measurement will include a margin for error. In fact, the results from spectroscopy measurements are not complete unless the error margins are provided with each piece of data. In general, the precision requirements in geochemical analysis depend on the particular geochemical contrast that is being investigated. The required level of precision must be sufficient for detecting subtle geochemical anomalies and variations, rather than an arbitrary fixed value [13]. Therefore, the target for analytical precision depends on the particular geochemical contrast that is being investigated and the analytical geochemists must be able to understand the required precision for an element at a particular concentration for a particular application (Table 2). The $\sigma$ value (the 'target value for standard deviation') for each element is in the higher precision for the 'pure geochemistry' type of analysis. Here, the analytical methods and protocols for geochemical analysis are designed with a lot of care for obtaining data of the highest accuracy, sometimes at the expense of a reduced sample throughput rate. Category 1 was calculated as $\sigma = 0.1c^{0.8495}$, where $\sigma$ and c (the analyte concentration) are expressed as fractions (i.e., $1\ \mu g/g \equiv 10^{-6}$). This specification is similar to the 'Horwitz function'. Category 2 is for laboratories working to an 'applied geochemistry' standard of performance, where, although precision and accuracy are still important, the main objective is to provide results on large numbers of samples collected, for example, as part of geochemical mapping projects or geochemical exploration program. For the lower precision 'applied geochemistry' type of analysis (Category 2), the $\sigma$ value was double the value of pure geochemical studies (Category 1). An appropriate category of uncertainty limits must be selected for the intended purpose [14,15]. The target standard deviation (Ha) for each element assessed was calculated from a modified form of the Horwitz function as follows:

$$H_a = k \cdot Xa^{0.8495}$$

where $X_a$ is the concentration of the element expressed as a fraction; the factor k = 0.01 for pure geochemistry laboratories and k = 0.02 for applied geochemistry laboratories [14].

**Table 2.** The precisions (Relative Standard Deviation, %RSD) required for major, minor, and trace elements for different applications in geochemical studies, after [14].

| Concentration of the Analyte | Category 1, RSD% | Category 2, RSD% |
|:---:|:---:|:---:|
| 100% | 1.0 | 2.0 |
| 10% | 1.4 | 2.8 |
| 1% | 2.0 | 4.0 |
| 1000 µg/g | 2.8 | 5.6 |
| 100 µg/g | 4.0 | 8.0 |
| 10 µg/g | 5.7 | 11.4 |
| 1 µg/g | 8.0 | 16.0 |
| 0.1 µg/g | 11.3 | 22.6 |

RSD = $100\sigma/c$ = relative standard deviation, calculated from $\sigma = 0.1\ C^{0.8495}$ (Category 1 data) or $\sigma = 0.2\ C^{0.8495}$ (Category 2 data).

*Z-Score Calculation*

    Z-score data must be calculated on their individual data obtained for reference materials used in the analytical programs based on the certified data for assessing the accuracy of the data using the following equation:

$$z = \frac{x - \mu}{\sigma}$$
$$\mu = \text{Mean}$$
$$\sigma = \text{Standard Deviation}$$

    (1)

$z$ = z-score, $x$ = assigned value, $\mu$ = value obtained for a particular element, $\sigma$ = the target value for standard deviation. For example: $\mu = 0.45$ µg/g is the value obtained in a laboratory for U, $x = 0.40$ (certified value or assigned value), $\sigma = 0.033$ (target SD), then $z = 1.51$ (z-score). So, the z-score value of U is within the $-2 < 1 < 2$ range, and hence considered satisfactory for pure geochemical studies.

*3.2. Data Integrity*

    Data integrity is the degree to which data are complete, consistent, accurate, trustworthy, and reliable [16]. The most important aspect of this is to maintain the performance and reproducibility of the analytical procedure when it is used in routine analysis. Recently an American scientist, Frances Arnold, who won the Nobel Prize for Chemistry in 2018 along with George P Smith and Gregory Winter, for their research on enzymes, published her subsequent paper on the enzymatic synthesis of beta-lactams in the journal 'Science' in May 2019. Later on, the paper was retracted, because the results reported were not reproducible, and she apologized to the scientific community. This is an excellent example of honesty in scientific reporting. If an experiment is a success, one would expect to obtain the same results every time it was conducted. Hence, the constancy or repeatability of the data is extremely important. Data integrity provides assurance that the analytical work in the laboratory from the sampling to the calculation has been carried out correctly and systematically, and the same results will be obtained if repeated in any laboratory across the world. Table 3 presents gold values (µg/g) in borehole samples obtained by different techniques from three different laboratories in India for a gold exploration study in Karnataka, India. It can be seen that the data are very much consistent, and the minor variations observed may be mainly due to the 'nugget effect' of gold.

**Table 3.** Gold values (µg/g) in borehole samples obtained by different techniques at different laboratories in India in a gold exploration study.

| S.No | Borehole Sample | MSPL, Hospet | NGRI, Hyderabad | | Private Lab, Bangalore |
|---|---|---|---|---|---|
| | | Pb-Fire Assay-AAS | MIBK-AAS | Pb-Fire Assay-AAS | Pb-Fire Assay-ICP-MS |
| 1 | S-101 | 3.20 | 3.17 | 4.34 | 3.06 |
| 2 | S-102 | 0.50 | 0.07 | 0.30 | 0.04 |
| 3 | S-103 | 1.30 | 1.34 | 1.68 | 1.08 |
| 4 | S-107 | 3.40 | 3.42 | 3.49 | 3.22 |
| 5 | S-108 | 1.40 | 1.42 | 1.44 | 1.33 |
| 6 | S-200 | 0.30 | 0.22 | 0.31 | 0.16 |
| 7 | S-201 | 4.80 | 5.11 | 5.44 | 4.95 |
| 8 | S-202 | 1.30 | 1.74 | 1.94 | 1.52 |
| 9 | S-203 | 0.10 | 0.03 | - | 0.04 |

*3.3. Requirement of QA and QC Protocols for Portable Instruments during Field Studies*

At present portable analytical instruments are increasingly being used in mineral exploration studies. Recent success in the discoveries of new mineral deposits using these portable techniques has allowed these techniques to become very popular. Laboratory-based instruments such as AAS, ICP-AES, MP-AES, WD-XRF, ICP-MS, and HR-ICP-MS are normally housed in stationary locations with controlled environments and temperatures, whereas portable instruments such as pXRF, pLIBS, and µRaman spectrometers are normally used in variable temperature conditions and environments, mostly during the field studies. pXRF can collect large amounts of multielement data rapidly directly in the field at a relatively low cost. However, many times, the absence of well-documented calibration processes including the use of CRMs raises concerns about a lack of internal consistency within the datasets obtained. Normally the results obtained in the field do not always match with those obtained on the same samples in the laboratory due to several reasons: (i) the technical limitations of portable instruments, (ii) the differences in the environment including dust, (iii) temperature differences (vi) differences in moisture content, measurement depth, and (v) differences in sample preparation (sample geometry, density, grain size, and moisture). In most cases, accurate results are obtained when the sample is homogeneous as laboratory sample preparation procedures ensure sample homogeneity and representativity. However, in field conditions, bringing sample homogeneity and representativity is rather difficult if not impossible even with grinding accessories available along with the portable instrument for use in the field. This aspect can be taken care of by making multiple measurements in a random fashion or in a gridded pattern directly on a rock outcrop in the field, thereby the accuracy of the measurement can be improved. In the case of XRF, it is a surface/near-surface technique that is assumed to penetrate down a few micrometers to several millimeters depending on the nature of the sample matrix, and only measures the portion of the sample directly in front of the window. On several occasions, comparable data can be obtained by pXRF and laboratory instruments like ICP-MS on the same samples for certain elements such as Sr and Ni provided that the data quality aspects are met (Figure 1).

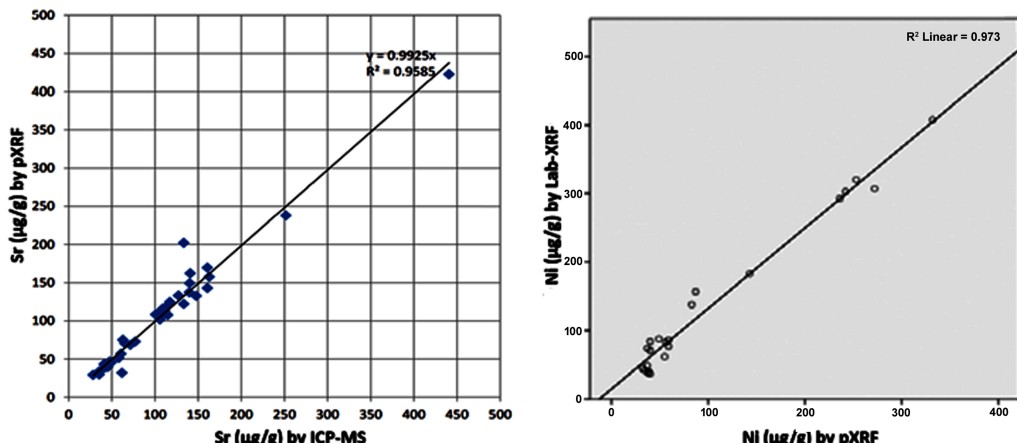

**Figure 1.** Favorable correlation between the data of strontium in a sandstone obtained by pXRF onsite in the field and by ICP-MS in a laboratory and Ni by pXRF in the field and XRF in a laboratory [17,18].

## 4. Contributions from the Recent Advances in Analytical Instrumentation

All potential users of geochemical data must have a clear understanding of the developments in modern analytical techniques in order to achieve their desired goals. A lack of familiarity with the geochemical analysis and the related analytical instrumentation could lead to the misapplication of some analytical techniques to various geochemical problems. Geochemical analysis is very challenging because of the complex compositions of different geological materials. First of all, the choice of analytical technique is of vital importance for generating meaningful data for the study under consideration. In fact, the choice of an analytical technique for a particular application is determined by the combination of several factors, including some basic analytical requirements such as (i) the nature of the sample, (ii) the analyte element, (iii) its concentration level, (iv) isotope information, (v) speciation information, (vi) instrument sensitivity/selectivity/resolution/portability, (vii) the level of accuracy required, (viii) the selection of appropriate sample digestion, (ix) throughput requirements, and (x) resources available and cost. It is not surprising that there are significant improvements in the quality of geochemical data, especially over the last half-century which occurred alongside progress in analytical methods and instrumentation. Especially the Apollo lunar sample return program gave a big push on the quality of geochemical data and minimized the required sample volumes. Those studies of rock and soil samples from the Moon yielded useful information about the early history of the Moon, the Earth, and the inner solar system [19]. In addition, the advent of ICP-MS in the early 1980s brought a remarkable improvement in the number of trace elements in geochemical materials as well as the quality [20,21]. Certainly, the geochemical analytical techniques have witnessed a powerful shift toward ICP-MS technology since 1983 when the first commercial ICP-MS instruments were released. Further instrumental developments in ICP-MS led to the development of HR-ICP-MS and MC-ICP-MS, and also the incorporation of a variety of interference removal technologies which had a greater impact on the quality of geochemical data (both elemental and isotopic), although several other analytical techniques including AAS, XRF, INAA, and ICP-OES are also available for geochemical analysis [22–24]. The newly developed MP-AES with its promising performance was also extensively utilized in recent years for the analysis of most complex geological materials such as rocks and ores and is becoming established slowly as a promising alternative analytical technique to AAS and ICP-AES for geochemical analysis [25,26]. In addition, developments such as ICP-MS/MS, for the effective removal of interferences using chemical resolution procedures, and several other useful features such as aerosol dilution (AD) in plasma-based analytical techniques permitting the nebulization of solutions of high TDS (<3–4%), with several advantages such as the minimization of matrix interference effects, dilution errors, liquid

waste, and sample contamination, have immensely contributed to the further improvement of quality [27].

*Microanalytical Techniques*

Earth scientists are also using microbeam analytical techniques such as an electron probe microanalyzer (EPMA), scanning electron microscopy energy dispersive spectroscopy (SEM-EDS), LA-ICP-MS, LA-ICP-MS/MS, LA-HR-ICP-MS, and LA-MC-ICP-MS, high resolution-SIMS (HR-SIMS) and SHRIMP for achieving accurate and precise element content, as well as isotopic ratio determinations even in single mineral grains [28–30]. These techniques combine the advantages of low detection limits with high spatial resolution. However, the accuracy and precision obtained on several occasions are restricted by many factors, such as sensitivity drift, elemental/isotopic fractionation, spectroscopic interferences, matrix effects, and the non-availability of sufficiently matrix-matched reference materials. Because of their large geometry and double-focusing abilities, both HR-SIMS and SHRIMP can offer resolutions of the order >5000R and can measure the isotopic and elemental abundances in minerals at a 10 to 30 $\mu$m-diameter scale and with a depth resolution of 1–5 $\mu$m [31,32]. In recent years, the chemical resolution capability of instruments such as LA-ICP-MS/MS has been playing a more significant role than mass resolution in the majority of the cases for the removal of interferences during chemical analysis [33,34].

There is also a definite shift of more and more measurements carried out on mineral grains than on whole-rock samples during the last two decades [35]. Microanalysis relies on the ability to obtain geochemical information from ever-smaller sample quantities translates into much-needed spatial and temporal resolution. Microanalysis is currently a basic technique for the dating of minerals, metamorphic events, and the understanding of mineral chemistry. Unlike conventional solution analysis, microanalytical techniques offer analysis with high spatial resolution, minimal sample preparation, low contamination, and high sample throughputs.

## 5. Contributions from the Recent Advances in Sample Preparation Techniques

Though there have been some significant advances in sample preparation methods using microwave digestion methods, infrared radiation digestions, high-pressure asher digestions, automated sample preparation systems (based on artificial intelligence), and the use of alternate reagents such as ammonium fluoride ($NH_4F$) and ammonium bifluoride ($NH_4HF_2$) instead of hazardous HF over the last three decades, the sample preparation remains one of the biggest challenges and limiting factors in the geochemical analysis [36–38]. Despite the fact that there have been several important developments in analytical techniques such as ICP-MS and HR-ICP-MS, the growth in the area of sample decompositions and digestion techniques does not commensurate with the innovations witnessed in analytical techniques. During international proficiency tests over the last 25 years, some of the recurring problems that surfaced were related to the dissolution of the refractory minerals such as xenotime, zircon, and chromite, which produce a large dispersion with consistent low values for Cr, Zr, Hf, and sometimes in HREE data obtained when acid digestion is involved [39]. Hence, there is a great need for the development of more effective sample decomposition methods to match the developments in analytical instrumental techniques. However, a variety of sample decomposition approaches are currently available to suit a variety of applications. For example, over forty samples can be digested in a single batch using the current commercially available microwave digestion systems. A high-pressure asher (HPA-S) can provide more effective digestions, taking the reaction pressure to 125 bar and the temperature to 300 °C [40]. This high-pressure digestion technique can effectively decompose refractory phases such as garnet, sphene, spinel, zircon, rutile, and chromite in a variety of geological materials and is a promising technique for both geochemical and geochronological studies. These capabilities of different sample digestion techniques can be fully exploited only when the merits and limitations of each method are completely

understood, and the most suitable sample preparation method and analytical technique can be utilized for a specific application.

## 6. Certified Reference Materials (CRMs)

During geochemical analyses, CRMs act as calibration standards, and also as 'controls' for checking the quality and metrological traceability of the data. They are particularly valuable in validating newly developed analytical methods, and analytical measurement procedures and protocols. Certified reference material is a particular form of measurement standard. The measurement of uncertainty in the geochemical analysis is used objectively to evaluate the quality of measurement results, which has a long history that goes back to the late 19th century, with serious measurement protocols developed in the early 20th century [41,42]. Due to the understanding of the importance of CRMs, their production, and development were started in a big way in the early 20th century itself [43,44]. An overview of the development of geochemical CRMs and the 137 years of their journey is presented in Table 4. Jochum et al. [45] and Jochum and Nohl [46] provided a new database named GeoReM (http://georem.mpch-mainz.gwdg.de accessed on 1 July 2022) for elemental, isotopic, and mineralogical geochemical standards which will be of immense value for geochemical, geochronological, and isotope studies. The above link provides all related information on geological CRMs.

**Table 4.** Chronology and a brief summary of geochemical CRMs over 137 years of journey.

| Year | Developments on CRMs/Major Contributors |
| --- | --- |
| 1885 | Six fertilizer samples were issued by the Association of Official Agricultural Chemist |
| 1901 | National Bureau of Standards (NBS) (it became the National Institute ofStandards and Technology (NIST) in 1988) |
| 1906 | This agency issued argillaceous limestone (as NBS-1) and zinc ore (as NBS-2). Other very early reference samples were copper and zinc ores, steel, and cement |
| 1966 | M. Roubalt, CRPG, France |
| 1973 | H.de la Roche and K Govindaraju, ANRT, France |
| 1976 | Flanagan, USGS, USA |
| 1982 | K. Govindaraju, ANRT, France |
| 1987 | A. Ando, GSJ, Japan |
| 1989 | X. Xie, IGGE, China |
| 1990 | S. Terashima, GSJ, Japan |
| 1994 | P.J. Potts, Open University, England |
| 2000s | GSI, NGRI, WIGH, NML, and NPL from India successfully prepared some geochemical CRMs |
| 2022 | Today, hundreds of rocks, ore, sediment, soil, and water CRMs are available with certified values (http://georem.mpch-mainz.gwdg.de/ accessed on 1 July 2022) for several elements and isotopes including specialized groups of elements such as REE and PGE from different agencies. |

GSI: Geological Survey of India; NGRI: National Geophysical Research Institute; WIGH: Wadia Institute of Himalayan Geology; NML: National Metallurgical Laboratory; NPL; National Physical Laboratory.

*Reference Materials for In Situ Isotopic Analysis*

Precise in situ measurements of certain isotopes and their ratios using microbeam analytical techniques on rock and mineral samples can provide valuable information on certain geochemical processes such as ore genesis, hydrothermal events, fluid-rock interaction, and melting processes. For example, clinopyroxene mineral is a major host for lithophile elements and can provide critical information on the mantle evolution and melt generation, and in situ $^{87}Sr/^{86}Sr$ isotopic analysis is valuable for such studies. Zhao et al. [47] attempted

to develop six natural clinopyroxene reference materials from South Africa (JJG1424) and China (YY09-47, YY09-04, YY09-24, YY12-01, and YY12-02) for Sr isotope microanalysis. The homogeneity of these potential reference materials was investigated and evaluated in detail over a 2-year period using 193 nm nanosecond and 257 nm femtosecond laser systems coupled with MC-ICP-MS and verified by the TIMS method. In addition, major and trace elements of these clinopyroxenes were determined by EPMA as well as solution and LA-ICP-MS. The new Sr isotope data will be useful in microbeam analysis applications.

## 7. International Proficiency Test Trials and Interlaboratory Comparisons

The International Association of Geoanalysts (IAG) has been organizing highly successful proficiency-testing (GeoPT) programs over the past 25 years for geochemical analysis [39,48]. Thompson [49] provided a detailed description of the objectives of international proficiency test trials and how they are conducted. The goal of interlaboratory exercises is to demonstrate the measurement capabilities of laboratories participating in interlaboratory comparisons (ILC) and proficiency tests (PT). These initiatives have certainly contributed to the improvement of the quality of the results and also an agreement between laboratories around the world that are involved in elemental analysis [50,51]. These exercises are designed to monitor and demonstrate the performance and analytical capabilities of the participating laboratories, and to identify gaps and problematic areas where further development is needed. The IAG conducted a whole-rock PT program for the first time in 1996 which involved 49 laboratories including the author's laboratory reporting concentration results on 51 elements from the Threlkeld microgranite of Cumbria (UK) [14,52].

## 8. Steps to Be Followed for Obtaining Accurate Data in Elemental and Isotopic Geochemical Studies

Geochemical research is a typically data-intensive discipline. At every stage of these investigations, a lot of care is required for the successful completion of the intended study. During the generation of analytical data, there are numerous sources of errors that may positively or negatively affect the results. In order to obtain reliable data in geochemical analysis, it is necessary to have an adequate awareness and understanding of possible sources of errors. The contamination of samples is not restricted to the laboratory alone; it can happen during all the processes of sampling, transport, and grinding, particularly during sample dissolution/preparation, and even during analysis. Apart from several others, gadgets used for sampling, sample containers during the fieldwork, grinding tools, and sample preparation methods during the analysis can also contribute. Geochemical field studies are most commonly taken up and designed to understand the compositional variations within bodies of rock, mineral, ore, soil, alluvium, and other geological materials. Collecting a representative sample is the most important task during fieldwork. Fundamentally there are two main types of errors—one associated with sampling in the field, and the other associated with chemical analysis in the laboratory. Errors of these types may significantly affect the interpretation and they can be minimized or controlled or even completely avoided, if possible, by taking suitable precautions in the field as well as in the laboratory. These aspects are discussed in a more detailed way in the following:

### 8.1. During Field Work

Geochemical sampling involves the collection of bedrock samples, soils, sediments, drill cores, and water. One must remember that correct interpretation depends only on the representativeness of the samples, and non-representative samples will not yield a valid interpretation no matter how good the subsequent chemical analysis will be. Hence, for the generation of meaningful and useful data, it is important to obtain representative samples while retaining their integrity during sample collection. It is always important to keep the aims of the study in mind and to determine what constitutes a representative sample of the material being investigated and the minimum mass or volume of each sample needed to meet the targets.

### 8.2. Sampling for Exploration Studies

Geochemical exploration studies require the analysis of large numbers of systematically collected samples to identify sources of geochemical anomalies for certain metals and delineate favorable areas for further exploration. Reliable sample collection during a mineral exploration program is very critical for its success. Accurate resource estimates depend on the collection of representative samples during exploration. Sampling programs must therefore be carefully designed to minimize the chances of collecting biased, unrepresentative, or contaminated material [53].

### 8.3. Sample Powder Preparation and Sample Homogeneity

In general, whole-rock samples are first reduced to mm-sized granules using a jaw crusher fitted with tungsten carbide plates to minimize contamination. An aliquot of 100 g is then reduced to a fine, homogeneous powder (<200 mesh) with a ball mill equipped with agate jars and milling balls. As shown in Table 5, grinding heads made of agate do not produce any measurable contamination when compared with other grinding tools and are the best for sample powder preparation.

**Table 5.** Trace elemental contamination by different grinding heads during the rock sample powdering process [54–57].

| Grinding Head | Contamination |
| --- | --- |
| Chrome-Steel | Fe, Cr, Mn |
| Tungsten Carbide | Co, Nb, Mo, W |
| Corundum | Al, Mg, Ba, Cu, Zn, Cr |
| High Carbon Steel | Fe, Cr, Zn, Mn, Cu, Ni |
| Alumina ceramic hand mill | Cs, W, Pb |
| The Fe hand mill | Mo, W |
| Agate Mill | No measurable contamination |

### 8.4. Choosing the Appropriate Amount of Sample for Analysis

Collecting the appropriate amount (mass or volume) of the sample is important, not only to lower the determination uncertainties with respect to the desired set of analytes, but also to ensure that there is enough sample to complete all required analyses, including checkups and loss on ignition in the case of solid samples, and keep a reserve split for any future use. Sample amounts required will be as high as 10 kg to obtain a representative sample for gold and PGE exploration studies, because of the heterogeneous distribution of these elements in crustal rocks [38]. Major, minor, and trace element analysis by WD-XRF normally requires 1–2 g samples for both pressed as well as fused pellets, and 50-100 mg sample amounts for trace elements (including REE) by ICP-MS and HR-ICP-MS [58,59].

### 8.5. Sample Digestion Methods for Analysis

Sample digestion in the geochemical analysis is particularly important because of the refractory nature of some of the geological materials. Rock samples are widely varied and contain refractory minerals such as chromite, zircon, barite, and rutile which are very difficult to dissolve completely. In ultramafic rocks, chromite is the main carrier of Cr, whereas zircon is the carrier mineral for REE, Zr, and HSFE. Recurring problems can be experienced with the dissolution of the refractory minerals such as zircon and chromite, which produce a large dispersion in data obtained particularly when acid digestion is involved. Clearly, no single sample digestion method can be applied to all types of samples to determine all elements. Hence, the sample digestion method chosen depends on the type of sample, the element/group of elements to be determined, and the analytical technique to be used for the determination.

### 8.6. Water Samples

Water samples are collected for a variety of studies related to hydrogeochemistry, marine geochemistry, mineral exploration, and the environment [60,61]. Although fresh and saline water are analyzed in different ways, the sample collection protocols are similar for both except for the fact that suspended particles are either filtered (<0.45 μm) or unfiltered depending on the type of investigation. Sample acidification by 0.1M high pure $HNO_3$ minimizes solute deposition, while the usage of HCl must be discouraged, considering $ArCl^+$ and $ClO^+$ interferences on $^{75}As^+$ and $^{51}V^+$, respectively, in ICP-MS measurements [62]. Acidifying the sample to a pH < 2 effectively stabilizes the trace element concentration for 180 days and refrigeration reduces potential biological activity [63].

### 8.7. Analysis Protocols

Quality control measures such as the use of procedural blanks, duplicates, and calibration standards and check samples (with known values) are important during analysis to ensure that the results produced are fit for purpose. In the case of water analysis by ICP-MS, the use of xenon isotope ($^{129}Xe$) is beneficial as it is already present as an impurity in the plasma gas. This will avoid external additions and will be of great value, particularly when determinations are carried out for elements present at ng/mL and sub-ng/mL levels [64].

### 8.8. Short and Long-Term Precision during Analysis

Monitoring the instrument signal and the precision of analysis with time is an important prerequisite and they are of two types, namely, short-term precision and long-term precision. Short-term precision is obtained from at least three replicate analyses of a sample based on which the relative standard deviation (% RSD) is also calculated. The RSD of less than 3% is considered good precision for concentrations varying in the range of 1000s of μg/g, while it can be less than 10% for those ultra-trace concentrations in sub-ng/g levels. The use of internal standardization is the proper remedy to this kind of problem [24].

### 8.9. Calibration Using Matrix-Matched CRM

Calibration curves are commonly generated in two ways: (i) using synthetic single or multi-element solutions of progressively increasing concentrations, and (ii) using matrix-matching CRMs with a reasonable spread in the analyte concentration. To verify the robustness of these calibration curves, a couple of CRMs can also be run as unknowns in the analytical sequence and check the data by comparing with certified values for different elements, and this comparison gives a direct measure of the accuracy of the method utilized. Figure 2 presents calibration curves generated using matrix-matched reference materials for the determination of gold by F-AAS and MP-AES work to minimize matrix interference effects [65,66].

### 8.10. Preparation of Duplicate/Replicate Samples

Splits of one sample taken after the coarse crush but before pulverizing (pulp) must be routinely used during sample preparation (usually 1 in 40 samples in exploration work). The quality control measures in the geochemical analysis must include the use of blanks, duplicates, and CRMs to ensure that the results produced are fit for purpose and to demonstrate that sampling and analytical variances are smaller relative to geological variance.

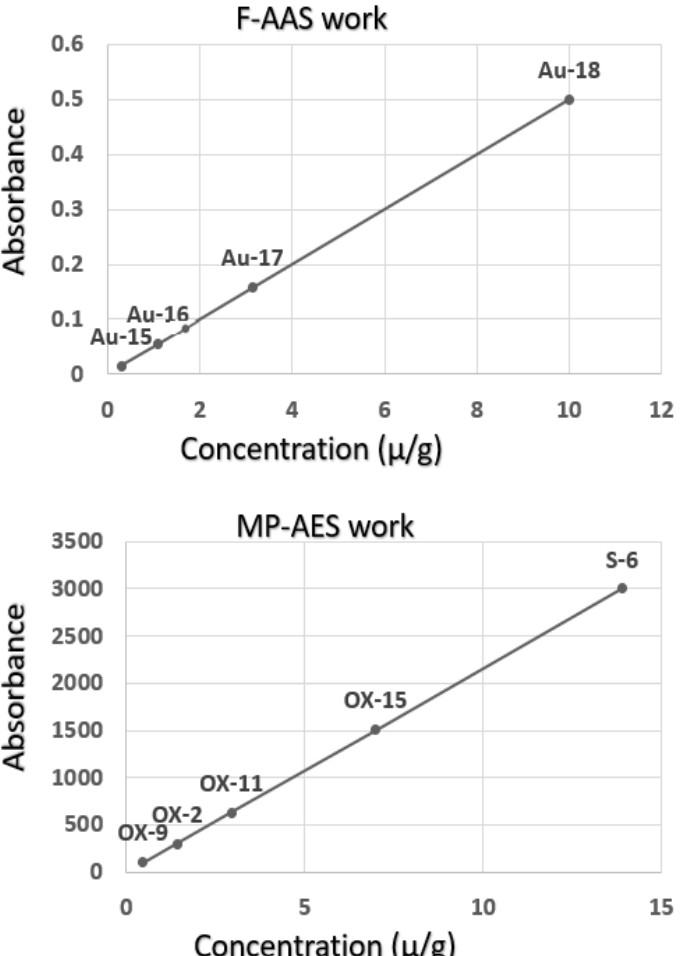

**Figure 2.** Calibration curves for Au with matrix-matching reference materials of different concentrations. Samples and calibration solutions were diluted in a similar way and the concentration calculation was straightforward [65,66].

*8.11. Setting Up a Sample Analysis Sequence*

Duplicates in geochemical investigations are of two types: field duplicates (splits of drill core or collected within a given distance from the original sample), and laboratory duplicates which are used to quantify the total sampling, and preparation precision and the laboratory duplicates to provide analytical precision. The analytical replicates and field duplicates are inserted to assess the overall precision of the analytical results individually for each element and to calculate the practical quantification limit (PQL) for each element analyzed. The procedural blanks are used to monitor contamination and to perform necessary corrections. Check samples or control samples are used to monitor accuracy. Figure 3 shows an example of the analytical sample sequence followed in our laboratory for the geochemical analysis by HR-ICP-MS [59]. By inserting CRMs in sample sets, the researcher can assess and establish the accuracy of the analytical results for the unknown samples.

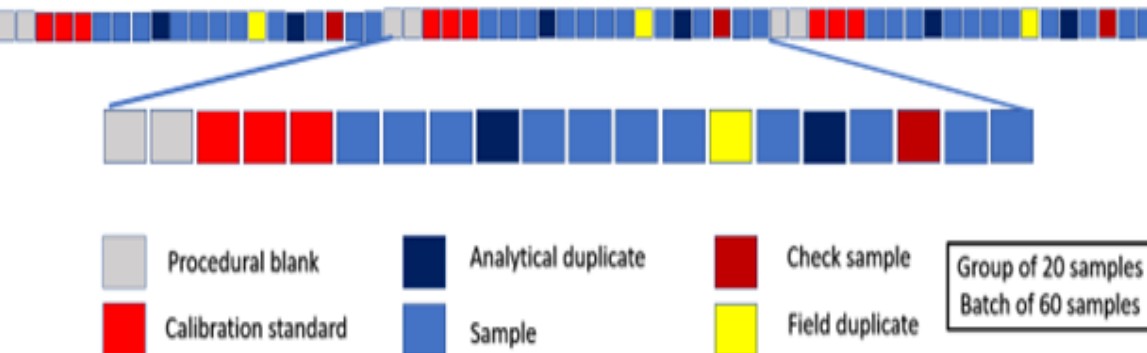

**Figure 3.** Typical analytical sequence of a batch of samples including duplicates, procedural, and calibration standards.

### 8.12. Procedural Blank Subtraction and Controlling Contamination

All analytical instruments will produce a small signal (background noise) even when the sample is not aspirated, or even when there is no analyte in the sample. Blank subtraction is critical for the analysis of both low- and high-abundance elements, hence, a reagent blank/procedural blank (prepared after using all the reagents) must also be analyzed, and 18.2 MΩ cm water must be used at various steps of the sample preparation procedure. Procedural blanks are useful in that they reveal background levels and possible contamination during the entire laboratory process. Sometimes, if the analyte signal only receives influence from the sample matrix and not that of the blank (without the sample matrix), the sample signal will be less than the blank signal, resulting in negative values for some elements. Multiple aliquots of procedural blanks and dilutions can be employed to minimize this type of analytical uncertainty [67]. Once total analytical blanks are at sufficiently low levels, they can be used to correct for elemental and isotopic concentrations in the unknown samples.

### 8.13. Intercomparisons with Different Techniques

It is very easy to check elemental concentrations or isotope ratios obtained by a particular technique by other techniques as there are a number of instrumental analytical techniques available today. If the sample dissolution (if applicable) and other analytical protocols are properly followed, one is expected to obtain the same result by any other analytical technique. For example, high-precision trace element data in NIST SRM 612 (trace elements in glass) obtained by ID-TIMS can be seen in comparison with those obtained by ICP-MS (Table 6). In another example, Zhu et al. [68] obtained REE concentrations in water reference material, SLR-4 by ICP-MS/MS, which are comparable with the values obtained by other well-established analytical techniques such as conventional single quadrupole ICP-MS, HR-ICP-MS, isotope dilution-HR-ICP-MS, along with certified values (Table 7).

### 8.14. Rejection of Outliers

Many statistical tools are available for the identification of outliers [74] (e.g., Dixon's test, Cochran's test, and Grubb's test were outlined in the ISO 5725 guide (2nd part)), where it was recommended to remove outliers from data sets in order to reach a higher degree of precision [75]. Fundamentally, the rejection of outliers requires that the average value is least affected.

**Table 6.** Concentrations (µg/g) of some trace elements by ID-TIMS in NIST SRM 612 (trace elements in glass) in comparison with those obtained by ICP-MS.

| Element | ID-TIMS [57] | ICP-MS [69] | ID-ICP-MS [57] |
| --- | --- | --- | --- |
| K | 60.63 ± 0.003 | - | |
| Rb | 31.79 ± 0.001 | 32 ± 2 | 31.07 ± 0.31 * |
| Sr | 78.36 ± 0.09 | | 78.51 ± 0.94 |
| Ba | 39.69 ± 0.01 | 36 ± 4 | 39.37 ± 0.47 |
| La | 35.85 ± 0.06 | | 34.65 ± 0.80 * |
| Ce | 38.73 ± 0.04 | 37 ± 3 | 37.25 ± 0.89 * |
| Nd | 35.95 ± 0.03 | 35 ± 4 | 34.96 ± O.63 |
| Sm | 38.07 ± 0.02 | 35 ± 3 | 37.15 ± 0.56 |
| Gd | 36.67 ± 0.19 | 36 ± 3 | 38.56 ± 0.66 |
| Dy | 36.28 ± 0.09 | - | 36.15 ± 0.65 |
| Er | 38.70 ± 0.10 | 36 ± 3 | 38.86 ± 0.43 |
| Yb | 39.16 ± 0.15 | 36 ± 3 | 40.14 ± 0.68 |
| Lu | 36.93 ± 0.05 | 35 ± 3 | - |

* Obtained by ICP-MS.

**Table 7.** The analytical results of REE in water reference material, SLRS-4 obtained by different well-established analytical techniques in comparison with those obtained by ICP-MS/MS.

| REE | Concentration (ng/mL) | | | | |
| --- | --- | --- | --- | --- | --- |
| | ICP-MS/MS [68] | ICP-MS [70] | HR-ICP-MS [71] | ID-HR-ICP-MS [72] | Compiled Value [73] |
| La | 294.5 ± 3.2 | 302.2 ± 7.3 | 279 ± 12 | 290.3 ± 6.4 | 287 ± 8 |
| Ce | 357.5 ± 3.2 | 378.4 ± 8.2 | 369 ± 15 | 364.1 ± 3.5 | 360 ± 12 |
| Pr | 70.9 ± 0.4 | 73.6 ± 1.5 | 75.4 ± 8.0 | 70.6 ± 2.3 | 69.3 ± 1.8 |
| Nd | 274.2 ± 3.2 | 277.4 ± 5.7 | 261 ± 9 | 270.3 ± 2.8 | 269 ± 14 |
| Sm | 58.5 ± 1.9 | 59.3 ± 1.4 | 54.3 ± 5.0 | 57.2 ± 0.3 | 57.4 ± 2.8 |
| Eu | 8.06 ± 0.41 | 8.09 ± 0.61 | 8.4 ± 0.8 | 8.00 ± 0.7 | 8.0 ± 0.6 |
| Gd | 33.86 ± 1.46 | 35.13 ± 1.01 | 38.3 ± 6.0 | 33.80 ± 0.36 | 34.2 ± 2.0 |
| Tb | 4.27 ± 0.20 | 4.50 ± 0.23 | 4.1 ± 0.5 | 4.30 ± 0.12 | 4.3 ± 0.4 |
| Dy | 22.82 ± 0.75 | 23.91 ± 0.66 | 21.7 ± 3.0 | 23.60 ± 0.16 | 24.2 ± 1.6 |
| Ho | 4.39 ± 0.19 | 4.86 ± 0.11 | 4.2 ± 0.5 | 4.60 ± 0.18 | 4.7 ± 0.3 |
| Er | 13.21 ± 0.46 | 13.53 ± 0.70 | 11.4 ± 3.0 | 13.10 ± 0.06 | 13.4 ± 0.6 |
| Tm | 1.75 ± 0.11 | 1.91 ± 0.04 | 1.8 ± 0.2 | 1.80 ± 0.02 | 1.7 ± 0.2 |
| Yb | 11.73 ± 0.36 | 12.03 ± 0.51 | 10.6 ± 2.0 | 12.30 ± 0.07 | 12.0 ± 0.4 |
| Lu | 1.76 ± 0.09 | 1.86 ± 0.11 | 1.7 ± 0.4 | 1.95 ± 0.02 | 1.9 ± 0.1 |

*8.15. Post-Laboratory Geochemical Analysis Quality Indicators*

Once the results are obtained from the laboratory, it is the responsibility of the scientist to review the results and check for discrepancies and QA/QC issues, discuss with the analyst at length, and solve the issues, if any, before starting interpretation. The comparison of the data with that obtained on the control samples will be useful to solve such issues many times. After the verification of the data, the same can be used for interpretation. There are also other post-laboratory analysis quality indicators. For certain rocks originating from a defined tectonic environment and of a certain age, certain element ratios (Zr/Hf = 36–37; Nb/Ta = 17–18; Y/Ho = 28) can be used as a quality indicator for the chemical analysis [76].

Deviation from these values may indicate interferences in the analytical process (like incomplete sample digestion and inaccurate calibration) or a different geochemical environment. However, at times, the deviations of Zr/Hf values from the said values may not indicate incomplete sample digestion, because, sometimes, even after the partial dissolution of the mineral zircon, the ratio of the Zr/Hf in the solution can be the same as in the mineral or in the rock. In addition, smooth REE and PGE chondrite-normalized curves also provide the additional assessment of the accuracy of the analysis.

*8.16. Data Processing and Analysis*

The application of statistical methods during the interpretation of geochemical data can be of considerable value to geochemists in both basic and applied studies. Many public domain and commercial packages are available for multi-element geochemistry data analysis methods (exploratory). Sampling and analytical precision are calculated using a procedure based on the analysis of variance (ANOVA). Spreadsheets have evolved over the years to incorporate complex formulas, macros, equations, different types of charts, etc., to ease systematic and routine statistical operations. Iwamori et al. [77] present a review of new statistical methods that effectively captures the structures of various types of multivariate geochemical data.

## 9. Conclusions and Future

The principle aim of this review is to highlight various coherent issues, such as the use of proper sampling protocols, avoiding contamination during sample grinding and cross-contamination, and the incomplete dissolution of the samples, in addition to spectroscopic and matrix effects faced during geochemical analysis, which can lead to the generation of inaccurate results. An analytical protocol containing the designing of a proper sample dissolution procedure and calibration using matrix-matching CRMs for obtaining accurate and precise results during geochemical analysis for both pure geochemical studies and mineral exploration studies is described. The use of CRMs for calibration will minimize the matrix and spectroscopic interference effects if not totally eliminated. The preparation of calibration standards using CRMs is easier, especially when the multi-element analysis is being carried out. In addition, CRMs also help in validation tests and quality checks and ensure that the results obtained are reliable and accurate.

Participation in international proficiency test trials and interlaboratory comparison exercises is extremely useful to understand the analytical protocols that are being adopted by other laboratories involved in similar studies. In addition, it is essential that the analyst must update the knowledge from time to time with the latest developments in all aspects of sampling, sample preparation, calibration methods, and newer analytical techniques to further improve the quality standards. A lot of developments are taking place in analytical technology, with the introduction of better methods for sample preparation, new pre-concentration techniques, and new types of equipment that are being introduced from time to time. Currently, there is a number of opportunities for the improvement of personnel training even. The measures outlined in this article are sufficient to ensure that geochemical data of appropriate quality are produced which can lead to unambiguous interpretations. The methodology described here includes the validation processes for obtaining reliable and reproducible data for extracting useful information to understand the various geological/geochemical processes of a particular lithology and is also useful in geological mapping and mineral exploration studies.

**Author Contributions:** Conceptualization, draft preparation, review and editing: V.B. and M.S. All authors have read and agreed to the published version of the manuscript.

**Funding:** This research did not receive any specific grant from funding agencies in the public, commercial, or not-for-profit sectors.

**Acknowledgments:** The authors thank the Director, CSIR-NGRI, Hyderabad, for the support and encouragement.

**Conflicts of Interest:** The authors declare no conflict of interest.

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
