# Peer review of "Data Quality in Geochemical Elemental and Isotopic Analysis"

_minerals, doi:10.3390/min12080999_

Round 1

Reviewer 1 Report

The manuscript deals with the very interesting subject of geochemical analyses and the processes that need to be followed in order to get both reliable and accurate results.

The authors covered all aspects from sampling to lab preparation and analyses.

Please see the attached version of the manuscript for details on the review

Author Response

Reviewer 1

All the comments made by reviewer 1 are relevant, and appropriate corrections have been made in the revised manuscript.

Reviewer 2 Report

Dear colleagues, authors of this paper submitted to MINERALS

This is certainly one of the best review articles that I have ever reviewed during the last two years. I positively have enjoyed and benefited from reading this outstanding monograph.

Best of luck in your future work.

Author Response

The authors would like to convey their heartfelt thanks to all three reviewers for their constructive comments and suggestions which definitely helped in further improvement of the quality as well as the focus of the manuscript.

Reviewer 3 Report

Balaram and Satyanarayanan provide an integrated review paper on the data quality in geochemical elemental and isotopic analyses. Generally, the topic of the manuscript is of international interest, the work although mostly theoretical and less experimental is important, the analysis and methodology are clear enough, and the manuscript is generally very well-structured. The figures are appropriated as both quantity and quality. The length of the paper, is appropriated for this journal, with all interpretations and conclusions to be in general well justified. The text is also very well organized, and this makes the manuscript easily readable and understandable. Finally the bibliography is accurate, without self-citations, and quite updated. The English is in relatively good shape, but some places need some improvements (please see my minor comments below). Overall, I have a couple of suggestions, and therefore ask for revision (minor) before accepting this manuscript for Minerals. So, please take them into account in order this promising contribution to be publishable. The manuscript is acceptable with very minor revision.

-L98-101: Rephrase it or split it into 2 different sentences

-L170-171: It is not clear at all. Please rephrase it

-In section 7, the Greaves et al. interlaboratory study could be added

M. Greaves, N. Caillon, H. Rebaubier, G. Bartoli, S. Bohaty, et al.. Interlaboratory comparison study of calibration standards for foraminiferal Mg/Ca thermometry. Geochemistry, Geophysics, Geosystems, AGU and the Geochemical Society, 2008, 9 (8), ï¿¿10.1029/2008GC001974ï¿¿.

Author Response

Reviewer 3

All the comments made by reviewer 3 are indeed helpful in increasing the analytical focus of the manuscript. The corrections were made as suggested by the reviewer. The spell check and the grammatical corrections were also made in the entire manuscript.

The references suggested by the reviewer have also been incorporated in section 7.

The authors take this opportunity to thank both the reviewers for the suggestions which have helped in improving the quality and depth of the manuscript.
